# SARS-CoV-2 serosurvey across multiple waves of the COVID-19 pandemic in New York City between 2020–2023

Juan Manuel Carreño[1,2,12], Abram L. Wagner [3,12], Brian Monahan [1,2],
Gagandeep Singh [1,2], Daniel Floda[4], Ana S. Gonzalez-Reiche [4],
Johnstone Tcheou[1,2], Ariel Raskin [1,2], Dominika Bielak[1,2], Sara Morris[1,2],
Miriam Fried[1,2], Temima Yellin [1,2], Leeba Sullivan[1,2], PARIS study group*,
Emilia Mia Sordillo [5] ✉, Aubree Gordon [3] ✉, Harm van Bakel [1,4,6,7] ✉,
Viviana Simon [1,2,5,8,9] ✉ & Florian Krammer [1,2,5,10] ✉

Sero-monitoring provides context to the epidemiology of severe acute respiratory syndrome coronavirus 2 (SARS-CoV-2) infections and changes in population immunity following vaccine introduction. Here, we describe results of a cross-sectional hospital-based study of anti-spike seroprevalence in New York City (NYC) from February 2020 to July 2022, and a follow-up period from August 2023 to October 2023. Samples from 55,092 individuals, spanning five epidemiological waves were analyzed. Prevalence ratios (PR) were obtained using Poisson regression. Anti-spike antibody levels increased gradually over the first two waves, with a sharp increase during the 3rd wave coinciding with SARS-CoV-2 vaccination in NYC resulting in seroprevalence levels >90% by July 2022. Our data provide insights into the dynamic changes in immunity occurring in a large and diverse metropolitan community faced with a new viral pathogen and reflects the patterns of antibody responses as the pandemic transitions into an endemic stage.

Coronavirus disease 2019 (COVID-19) has severely impacted human health worldwide. Within the United States, New York City emerged as an early epicenter of infection and cases, with the first reported case on February 29, 2020[1]. However, public reporting of cases over time was inconsistent due to limited testing capacity early in the pandemic[1]. Moreover, testing was differentially available to groups depending on their socioeconomic status[2], impacting understanding of the course of the epidemic in different populations.

Severe acute respiratory syndrome coronavirus 2 (SARS-CoV-2), the causative agent of COVID-19, has rapidly evolved to increase its transmission potential and to evade immunity derived from prior infection and vaccination, contributing to several waves of infections[3].

[1]Department of Microbiology, Icahn School of Medicine at Mount Sinai, New York, NY, USA. [2]Center for Vaccine Research and Pandemic Preparedness (C-VaRPP), Icahn School of Medicine at Mount Sinai, New York, NY, USA. [3]Department of Epidemiology, School of Public Health, University of Michigan, 1415 Washington Heights, Ann Arbor, MI 48109, USA. [4]Department of Genetics and Genomic Sciences, ISMMS, New York, NY, USA. [5]Department of Pathology, Molecular and Cell Based Medicine, Icahn School of Medicine at Mount Sinai, New York, NY, USA. [6]Icahn Genomics Institute, ISMMS, New York, NY, USA. [7]Department of Artificial Intelligence And Human Health, Icahn School of Medicine at Mount Sinai, New York, NY, USA. [8]Division of Infectious Diseases, Department of Medicine, Icahn School of Medicine at Mount Sinai, New York, NY, USA. [9]The Global Health and Emerging Pathogens Institute, Icahn School of Medicine at Mount Sinai, New York, NY, USA. [10]Ignaz Semmelweis Institute, Interuniversity Institute for Infection Research, Medical University of Vienna, Vienna, Austria. [12]These authors contributed equally: Juan Manuel Carreño, Abram L. Wagner. *A list of authors and their affiliations appears at the end of the paper. ✉e-mail: Emilia.sordillo@mountsinai.org; gordonal@umich.edu; harm.vanbakel@mssm.edu; viviana.simon@mssm.edu; florian.krammer@mssm.edu

The emergence of viral variants capable of overcoming pre-established immune responses is mainly attributed to mutations in the spike surface glycoprotein. During the first wave of infections, the wild-type (WT) virus acquired the D614G mutation that increased virus infectivity and transmissibility[4,5]. These strains prevailed for several months until the second wave of infections caused by the Alpha, Beta, and Gamma variants occurred in different parts of the world. A 3rd wave of infections was caused by the Delta variant which showed higher transmissibility and enhanced severe disease. A significant reduction of viral neutralization by sera from convalescent or vaccinated individuals was detected[6,7]. In December 2021, the emergence of the Omicron BA.1 strain, which carried over 30 spike protein mutations, resulted in a 4th wave of infections. Since then, distinct Omicron sublineages have circulated worldwide, including BA.2, BA.5, BQ.1, BQ.1.1, BF.7, and more recently XBB.1, XBB.1.5, BA.2.86, EG.5.1, JN.1, and derived variants.

Throughout the pandemic, monitoring the level of immunity in the population has been crucial for public health. Sero-monitoring has guided safety measures and vaccination policies worldwide to reduce viral infection, transmission, and severe disease[8]. Laboratories can measure binding antibodies to the spike (S) protein, which indicates past infection or vaccination, or antibodies to the nucleoprotein (NP), which indicates past infection(s). The level of anti-SARS-CoV-2 antibodies and T-cell responses correlate with an individual's capacity to combat the virus and with protection against infection and disease[9,10]. Importantly, binding antibody levels correlate with the number of prior exposures, and with the level of neutralizing antibodies[11,12].

Hospital-based sero-monitoring for SARS-CoV-2 provides an estimate of the level of immunity in the population and could provide a blue-print of how to conduct such surveillance during a future pandemic. This sero-monitoring could provide important information about the course of the pandemic, including who are high-risk groups, and the persistence of disparities even after the introduction of population-level interventions. Existing case reporting systems for COVID-19 could be limited by changing testing availability and behaviors over time. Here, we report the results of a prospective, hospital-based cross-sectional study of anti-S and anti-NP sero-monitoring from the beginning of February 2020 through July 2022, and follow-up measurements in August-October 2023. We describe the longitudinal seroprevalence in two different groups: (a) an 'urgent care' group that was enriched for acute cases of COVID-19 early during the pandemic, and (b) a 'routine care' group, resembling the general population, as previously described[8]. Stratification by zip code allowed us to study the geographical distribution of the seroprevalence and antibody titers in residents of NYC throughout the pandemic.

## Results

Overall, there were 55,092 individuals sampled between February 9, 2020 and July 18, 2022: 21,075 from urgent care group and 34,017 from the routine care group. Of these 882 were children <18, and 54,210 were adults. The demographic distribution among adults is shown in Table 1. In the routine care groups, a plurality of individuals were in the 18–44 year age group (16,499, 49%), a majority female (23,166, 69%), a plurality self-identified as White (14,971, 45%), and about half had private insurance (16,386, 49%). The distribution of these variables for children is shown in Supplementary Table 1. COVID-19 vaccination began in January 2021 in the general population and varied across a number of groups, including race/ethnicity. Uptake was faster in Asian and Pacific Islanders and White New Yorkers compared to Black New Yorkers (e.g., at wave 3, uptake in these three groups was 63%, 55%, and 50%, respectively) (Supplementary Table 2).

We measured spike antibody prevalence and titers over five epidemiological waves based on average case counts (7-day) of SARS-CoV-2 infections in New York City, reported by the NYC Department of Health and Mental Hygiene (NYC DOHMH) (Fig. 1A). Demographic

characteristics did not substantially vary by wave (Supplementary Table 3). Viral variant prevalence through this period continuously changed, from the initial circulation of ancestral variants to the appearance of variants of the Omicron lineage (Fig. 1B). Overall, among adults in the routine care group, spike protein seropositivity increased from an average of 13% in wave 1, to 36%, 76%, 88%, and 93% in the following four waves (Fig. 1C, D; Supplementary Table 4), consistent with seroprevalence estimates reported by the NYC Department of Health and Mental Hygiene (DOHMH)[13]. A sharp increase in antibody prevalence was detected during waves 2 and 3, starting in February 2021. This increase corresponded temporally with the expanded availability of SARS-CoV-2 vaccinations, the emergence of the Iota, Alpha, and Delta variants, and changes in public health measures. Importantly, antibody prevalence was maintained during the follow-up period from August 28st 2023 to October 2nd 2023 (Fig. 1C, D). Vaccinated individuals showed significantly higher antibody prevalence than non-vaccinated as early as January 2021, with the largest differences just prior to and during wave 3 (Fig. 2G, H) owing to incomplete vaccination in the earlier periods and unvaccinated individuals experiencing high infection rates during waves 2 and 3. By wave 5, spike protein seropositivity was uniformly high (≥90% in all categories). Similar trends were observed in populations stratified by sex and age, with no significant differences among intra-group strata (Fig. 2A–D). Differences by race/ethnicity were observed, for example, 14% of Black New Yorkers and 18% of those within the Other racial category were seropositive in wave 1, versus 7% of Asians and Pacific Islanders (Fig. 2E, F).

A sensitivity analysis of trends in seropositivity over time, using data weighted to Census 2022 age and gender estimates, did not reveal any substantively different results (Supplementary Figs. 1–2).

The multivariable Poisson regression model of spike protein seropositivity reveals several trends within demographic group and across time (Table 2, Supplementary Table 5). At wave 1, seropositivity was higher in younger age groups compared to older (1.66 times higher in those 18–44 vs 65+, 95% CI: 1.41, 1.96). By wave 5, this had largely attenuated (interaction $p$ value < 0.0001). Larger disparities by race/ethnicity were also observed at the beginning of the pandemic. Compared to White New Yorkers, Black (prevalence ratios (PR): 1.24, 95% CI: 0.94, 1.63) and Other New Yorkers (PR: 1.64, 95% CI: 1.44, 1.88) had higher seropositivity, and Asians and Pacific Islanders had relatively low seropositivity (PR: 0.67, 95% CI: 0.51, 0.87). During waves 3 and 4, some of these associations reversed, with lower seropositivity in Black and Other groups compared to White New Yorkers. Around a similar or earlier time (waves 2 and 3), vaccination coverage was relatively low among Black compared to White New Yorkers (Supplementary Table 2). An analysis of spike protein seropositivity across racial ethnic groups stratified by vaccination status reveals significant differences among the unvaccinated (Supplementary Tables 6 and 7), with seropositivity relatively high in waves 1 through 2 and then relatively low in waves 3 and 4 in Black New Yorkers. However, among the vaccinated, seropositivity remained equivalent across racial groups. Indeed, in the models with vaccination status that cover periods once the vaccine had been rolled out (Supplementary Tables 8 and 9), this factor emerges as a significant contributor to seropositivity, with a larger impact in wave 2, where seropositivity was 1.95 times higher among those vaccinated compared to unvaccinated (95% CI: 1.87, 2.03), while it was only 1.04 times higher in wave 5 (95% CI: 1.02, 1.07) (interaction $p$ value < 0.0001). Similar demographic patterns were shown in sensitivity analyses limited to adults 18–64 (Supplementary Tables 10 and 11) and those vaccinated (Supplementary Tables 12 and 13).

Anti-spike antibody titers in seropositive individuals were induced at moderate to high levels during the first waves of infections, with modest antibody decay over time (Fig. 1E, F; Supplementary Fig. 3). Amid the second wave, titers increased gradually reaching the highest

**Table 1 | Distribution of demographic characteristics in a SARS-CoV-2 serosurveillance system, New York City (N = 54,210)**

| | | In-patient departments N = 20,721 | Routine care departments N = 33,489 |
|---|---|---|---|
| Wave | Wave 1 (through 30 Aug 2020) | 5233 (25%) | 8634 (26%) |
| | Wave 2 (through 20 June 2021) | 7624 (37%) | 13562 (41%) |
| | Wave 3 (through 31 Oct 2021) | 3119 (15%) | 5359 (16%) |
| | Wave 4 (through 6 March 2022) | 2380 (11%) | 2849 (9%) |
| | Wave 5 (through 18 July 2022) | 2365 (11%) | 3085 (9%) |
| Age | 18–44 years | 4917 (24%) | 16499 (49%) |
| | 45–64 years | 6877 (33%) | 8155 (24%) |
| | 65+ years | 8927 (43%) | 8835 (26%) |
| Sex[a] | Female | 10183 (49%) | 23166 (69%) |
| | Male | 10535 (51%) | 10322 (31%) |
| Race/ethnicity | Asian Pacific Islander | 1201 (6%) | 3249 (10%) |
| | Black | 2934 (14%) | 2958 (9%) |
| | white | 6116 (30%) | 14971 (45%) |
| | Other | 7147 (34%) | 8416 (25%) |
| | Unknown | 3323 (16%) | 3895 (12%) |
| Insured | Private | 5902 (28%) | 16386 (49%) |
| | Public | 12781 (62%) | 14824 (44%) |
| | Other | 2038 (10%) | 2279 (7%) |
| Vaccinated | No | 14604 (70%) | 24584 (73%) |
| | Yes | 6117 (30%) | 8905 (27%) |

[a]One missing value for sex in the routine care departments group during wave 1.

levels during wave 5. No significant differences were detected when titers were stratified by gender, age, or race/ethnicity (Supplementary Fig. 4A–F). As expected, the introduction of vaccination during wave 2 resulted in increased antibody prevalence and titers at the time of sample collection, while individuals without a record of vaccination displayed significantly lower antibody prevalence through waves 2, 3, and 4 (Fig. 2 and Supplementary Fig. 4G, H). These differences narrowed over time, likely due to multiple infections in unvaccinated individuals. Moreover, differences in antibody prevalence and titers in vaccinated vs non-vaccinated individuals were preserved when stratification was based on sex, age, and race/ethnicity (Supplementary Figs. 5–10).

For select weeks (July 6–20th 2020, February 15th to March 1st 2021, June 1–14th 2021, August 16–30th 2021, May 23–30th 2022, August 21st to Oct 2nd 2023) we also measured seropositivity and titers for NP. Although initially in 2020 NP seropositivity was similar to the spike seropositivity, this changed in 2021 as NP seropositivity fell behind spike positivity by a relatively large margin (Fig. 3). This was influenced not only by vaccination (which only induces anti-spike antibodies) but likely also by waning of NP antibodies over time and the low induction of NP antibodies during breakthrough infections. Our estimates during mid-2021 (June 1–14th, 57% (95% CI = 53.1–60.9) to late 2021 (August 16–30th, 54% (95% CI = 50.5–57.5)) are higher than the reported nationwide NP seroprevalence (December 2021 33.5% (95% CI = 33.1–34.0))[14]. These differences may be due to discrepancies in the time points of antibody measurements. Antibodies induced or boosted during the Delta wave may have waned over the subsequent months, due to the relatively fast decay rates of NP antibodies induced by infection[11]. Alternatively, differences in the sensitivity of the methods used or higher COVID-19 incidence in NYC compared to nationwide levels[15] could have played a role.

To analyze the geographical distribution of antibody prevalence and titers through the five boroughs of NYC (Manhattan, Brooklyn, Queens, The Bronx, and Staten Island), we used neighborhood tabulation areas (NTAs) to build maps of the city divided into 216 NTAs (Supplementary Fig. 11). We detected higher antibody prevalence in

areas located outside Manhattan during all waves of infections, that were more evident during waves 1–3, when antibody prevalence was lower (Supplementary Fig. 11A–C). Importantly, seroprevalence reached high levels (above 80%) almost homogenously throughout the five boroughs by wave 5 (Supplementary Fig. 11E). Likewise, higher titers were detected in boroughs other than Manhattan (Fig. 4), with areas reaching the highest titers located in Queens (Fig. 4E).

## Discussion

Serosurveillance systems provide means to describe population patterns in infection and vaccination over time. Overall, in a hospital-based study in New York City, we previously described a sharp increase in anti-spike antibody prevalence early during 2020 due to SARS-CoV-2 infections[8]. By May 2021, seroprevalence reached levels above 70% due to both infections and vaccination programs. Here we report on the continued sero-monitoring efforts to capture antibody prevalence and titers over the course of the pandemic. We show that antibody prevalence gradually increased over time, reaching levels above 90% by January 2022. Seroprevalence and titers were maintained over time, with similar trends observed during a follow-up period in August–October, 2023, likely due to constant antigenic exposures by vaccine boosters and breakthrough infections. Moreover, we analyzed the geographical distribution of antibody prevalence and titers through the five boroughs of NYC. Finally, we analyzed NP titers–indicative of immunity derived by infection–at discreet time points during the pandemic.

Our seroprevalence estimates, for the most part, overlap with estimates from the NYC DOHMH. However, there are differences during the early phase of the pandemic and later during waves 4 and 5. Specifically, the NYC DOHMH reports a large proportion (>50%) of seropositive individuals in wave 1 who were tested for SARS-CoV-2 by PCR. Individuals seeking testing were probably more likely to be infected and potentially already antibody positive given the long initial incubation time[1]. This is similar to what we observed in the urgent care group that was biased towards SARS-CoV-2 positives in the first wave. In more recent waves, seropositivity is lower, according to the NYC

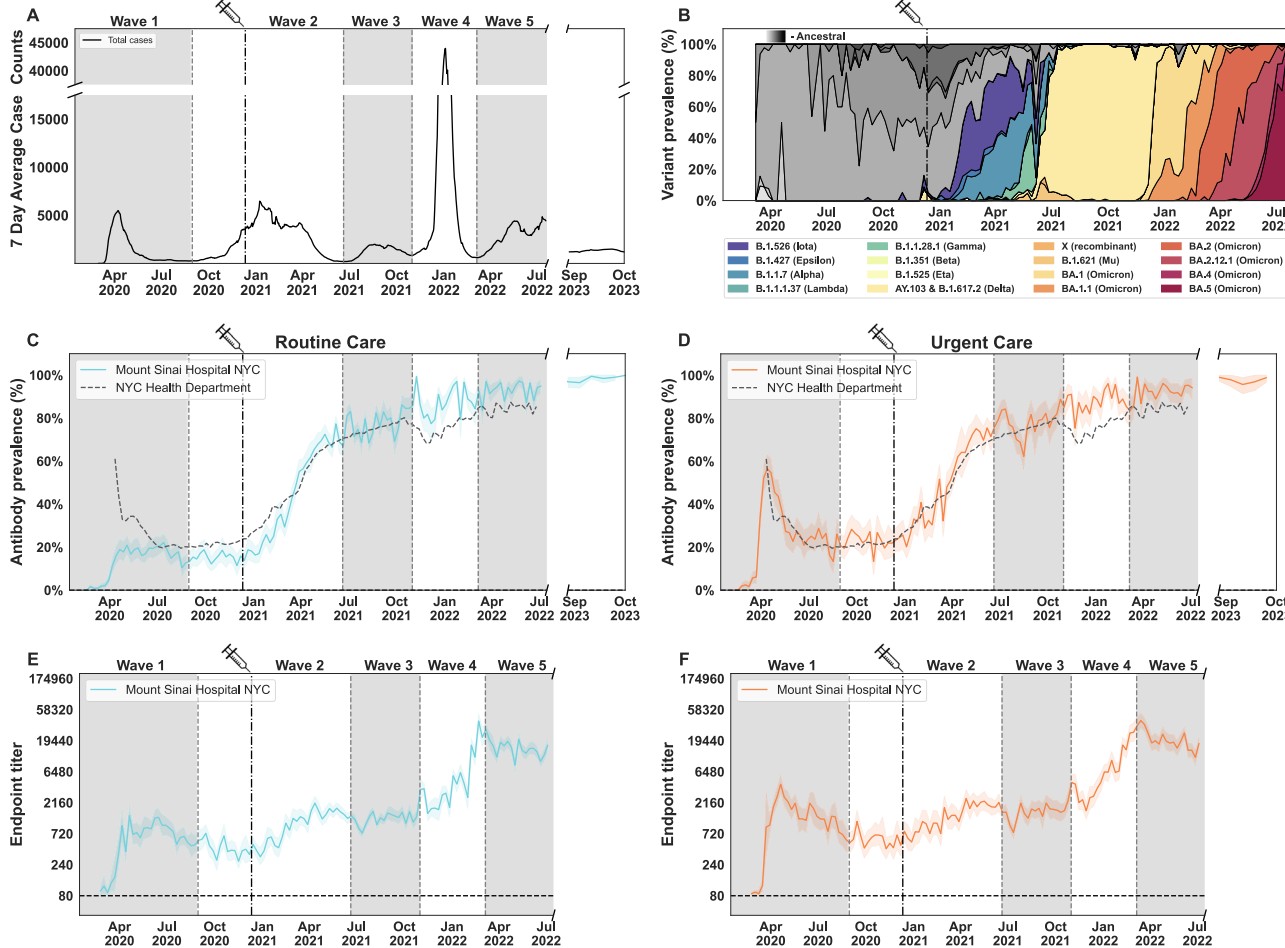

**Fig. 1 | SARS-CoV-2 spike binding antibody prevalence and titers in the Routine care and Urgent care groups at Mount Sinai Hospital in New York City (NYC).** 7-day rolling average case counts of SARS-CoV-2 infections in New York City reported by the NYC Department of Health and Mental Hygiene (NYC DOHMH) (**A**). Main circulating SARS-CoV-2 lineages presented as percent prevalence based on data from the Pathogen Surveillance Program at Mount Sinai (**B**). Mean antibody prevalence with 95% bootstrapped confidence intervals between February 9th 2020 to July 18th 2022 (**C**, **D**) and geometric mean antibody titers with boot-strapped 95% confidence intervals (**E**, **F**) are shown for the Routine Care (left column, blue lines) and Urgent Care (right column, orange lines) groups (February 9th 2020 to July 18th 2022 and August 21st 2023 to October 2nd 2023). Samples were run in duplicate in a 2-step ELISA protocol. All bootstrapped confidence intervals are based on random resampling of results using the above mentioned mean for each graph. **B** lineages before the emergence of variants of concern (VOCs) are shown in greyscale, while VOCs are shown in color. NYC DOHMH mean antibody prevalence data are shown in **C**, **D** for reference, denoted with a dashed line. The date on which the first FDA-authorized SARS-CoV-2 vaccine became available in NYC is indicated by the vertical dotted line and syringe. Alternating shaded areas in **A**, **C**–**F** denote the five successive epidemiological waves of infection in NYC. Gray color and vertical dashed lines serve as visual contrast.

DOHMH, than in our serosurveillance study. This could reflect differences in the assays used or differences in the study populations.

The COVID-19 pandemic intensified racial and economic disparities in health[16]. Our data collected in the early waves, indicate that seropositivity was relatively high in adults self-identifying as Black and Others, pointing to higher infection rates in essential workers lacking the possibility to socially distance or to socioeconomic constraints leading to more crowded living conditions[17,18]. Our serosurveillance system revealed other sociodemographic trends by age. These reflect epidemiological risk patterns and vaccination policies. For example, early in the pandemic, seropositivity was greater in younger adults, in accordance with risk messaging directed towards older adults. As vaccination became available and was prioritized for older ages[19], seropositivity became relatively higher in older age groups.

The spike antibody response induced towards the emerging Omicron sub-lineages is mostly cross-reactive with ancestral strains[20], hence the seroprevalence estimates presented here are likely to be preserved towards the spike of Omicron sub-variants. Although the neutralization capacity of sera from individuals exposed to ancestral

SARS-CoV-2 antigens through infection or vaccination is severely reduced against Omicron strains[21], non-neutralizing antibodies may protect through alternative mechanisms mediated by Fc-dependent antibody effector functions[22] and the presence of antibodies likely also indicates the presence of T-cell responses which largely cross-react between different variants of concern[23]. Dissecting the contribution of non-neutralizing cross-reactive antibodies upon exposure to emerging variants of the Omicron lineage will be important to have a broader picture of an individual's immunity.

Several sero-monitoring programs have been implemented worldwide in response to the SARS-CoV-2 pandemic. These systems have been designed for various purposes, including to estimate antibody seropositivity[24], to monitor development of community immunity[25], and to examine occupational risks[26]. Community-based seroprevalence studies (e.g. Kshatri et al.[27]) have several limitations, including costs of conducting field studies and non-response, which could be differential based on important demographic groups. Hospital-based sero-monitoring systems, as we describe, can leverage existing laboratory infrastructure in a low-cost manner.

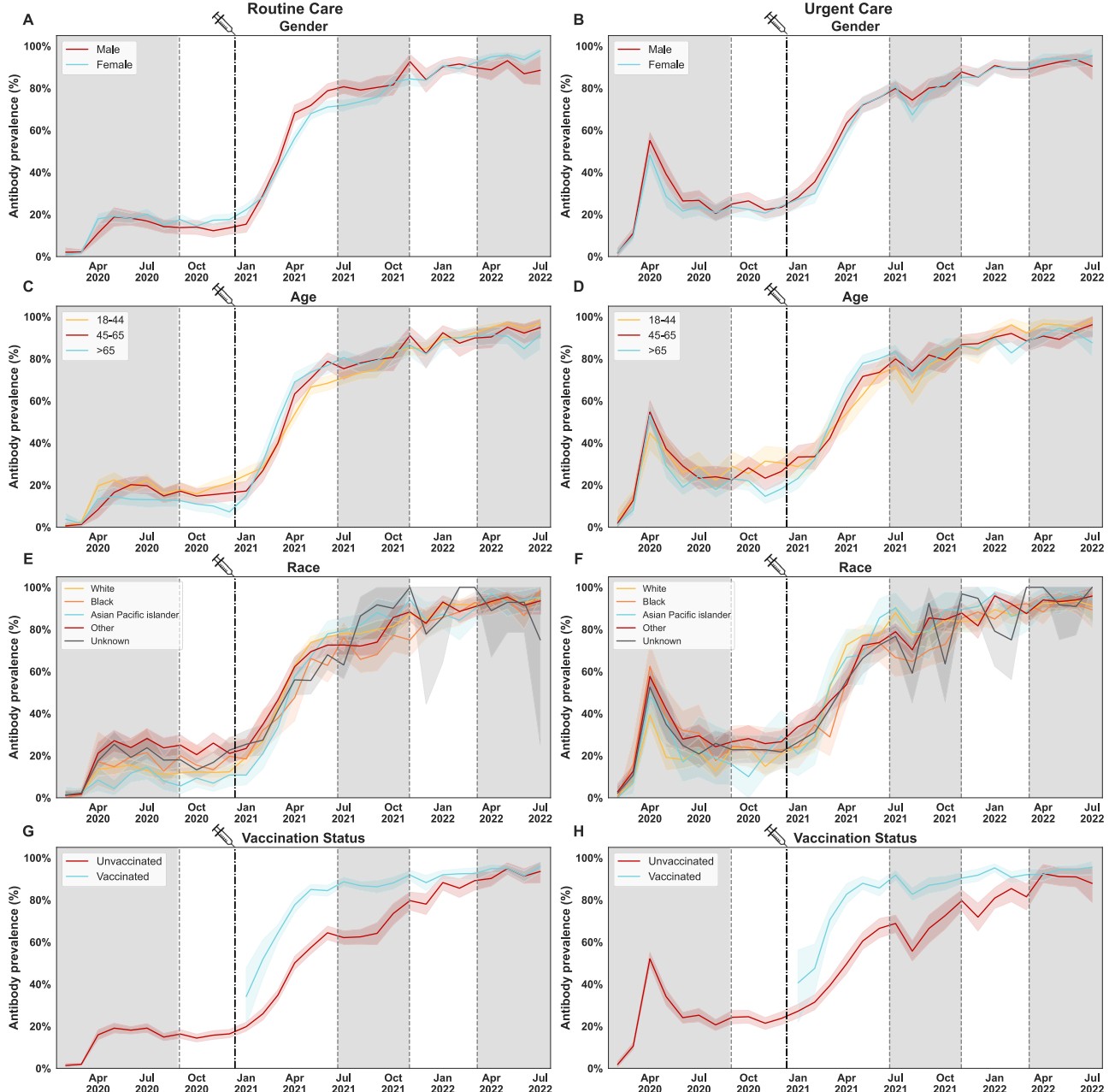

**Fig. 2 | SARS-CoV-2 spike antibody prevalence stratified by demographic groups and vaccination status.** Mean antibody prevalence in the Routine Care (left column), and Urgent Care (right column) groups, measured between February 9th 2020 to July 18th 2022, and stratified by gender (**A**, **B**), age (**C**, **D**), race (**E**, **F**), and vaccination status at the time of sample collection (**G**, **H**) is shown. Gray color serves as visual contrast. Vertical dashed lines separate waves of SARS-CoV-2 infection in NYC. Samples were run in duplicate in a 2-step ELISA protocol. The mean of each graph is paired with a bootstrapped 95% confidence interval based on random resampling of results. **A**, **B** Gender stratification: males, females.

**C**, **D** Categorical age levels: 18–44, 45–65, >65. Individuals <17 are not shown in this analysis. **E**, **F** Race/ethnicity stratification: White, Black, Asian, and Pacific Islander, Other, unknown. **G**, **H** The date on which the first FDA-authorized SARS-CoV-2 vaccine became available in NYC is indicated by the vertical doted line and syringe. Vaccination status was assessed at the time of sample collection and does not reflect vaccination rates in NYC or within our patient population. Alternating shaded areas in all graphs denote the five successive epidemiological waves of infection in NYC. Gray color and vertical dashed lines serve as visual contrast.

Our analysis has some limitations. Our catchment area was tied to individuals visiting the Mount Sinai Health System, and we may have limited scope to generalize outside the NYC metropolitan area. Residual, de-identified samples were collected, which impedes following individuals longitudinally. The representation of samples by different NTAs was limited, which limited the geographical distribution analysis.

New York City was an early epicenter of the COVID-19 pandemic. Leveraging a hospital-based serosurveillance system, we detect several patterns in line with the epidemiology of SARS-CoV-2 and the roll-out of the COVID-19 vaccines. Notably, by early 2022, virtually the entire population had some evidence of COVID-19 seropositivity. Importantly our data clearly show increasing overall seropositivity in the general population as SARS-CoV-2 vaccines became available. The overall sustained high level of implied immunity in the total population is also noteworthy. Finally, we believe our data set provides a great resource for additional modeling studies and could help to predict immunity and seroprevalence scenarios during future pandemics.

**Table 2 | Multivariable Poisson regression of spike protein seropositivity among those receiving routine care in a SARS-CoV-2 serosurveillance system, New York City (N = 33,488)**

| | Wave 1 (through 30 Aug 2020) PR (95% CI) | Wave 2 (through 20 June 2021) PR (95% CI) | Wave 3 (through 31 Oct 2021) PR (95% CI) | Wave 4 (through 6 March 2022) PR (95% CI) | Wave 5 (through 18 July 2022) PR (95% CI) | p value[a] |
|---|---|---|---|---|---|---|
| Age | | | | | | <0.0001 |
| 18–44 years | 1.66 (1.41, 1.96) | 0.94 (0.88, 1.00) | 0.93 (0.89, 0.97) | 1.01 (0.97, 1.05) | 1.05 (1.02, 1.08) | |
| 45–64 years | 1.30 (1.09, 1.55) | 0.98 (0.91, 1.04) | 0.98 (0.94, 1.02) | 1.02 (0.98, 1.07) | 1.04 (1.00, 1.07) | |
| 65+ years | ref. | ref. | ref. | ref. | ref. | |
| Sex | | | | | | 0.0001 |
| Female | ref. | ref. | ref. | ref. | ref. | |
| Male | 1.04 (0.91, 1.19) | 1.03 (0.98, 1.09) | 1.06 (1.02, 1.10) | 1.03 (0.99, 1.06) | 0.96 (0.94, 0.99) | |
| Race/ethnicity | | | | | | <0.0001 |
| Asian Pacific Islander | 0.67 (0.51, 0.87) | 0.88 (0.80, 0.96) | 1.05 (1.00, 1.09) | 0.99 (0.94, 1.04) | 1.01 (0.98, 1.04) | |
| Black | 1.24 (0.94, 1.63) | 1.08 (0.99, 1.18) | 0.89 (0.84, 0.94) | 0.93 (0.89, 0.98) | 0.99 (0.96, 1.02) | |
| White | ref. | ref. | ref. | ref. | ref. | |
| Other | 1.64 (1.44, 1.88) | 1.14 (1.08, 1.21) | 0.96 (0.93, 1.00) | 0.99 (0.96, 1.02) | 1.00 (0.97, 1.02) | |
| Unknown | 1.33 (1.14, 1.56) | 0.92 (0.86, 0.99) | 0.92 (0.86, 0.98) | 1.02 (0.93, 1.13) | 0.97 (0.89, 1.05) | |
| Insured | | | | | | 0.0018 |
| Private | ref. | ref. | ref. | ref. | ref. | |
| Public | 1.14 (1.00, 1.29) | 0.96 (0.91, 1.02) | 0.93 (0.90, 0.96) | 0.99 (0.96, 1.03) | 1.01 (0.99, 1.04) | |
| Other | 0.88 (0.70, 1.11) | 1.01 (0.92, 1.11) | 0.98 (0.92, 1.04) | 0.99 (0.94, 1.05) | 1.01 (0.97, 1.04) | |

*PR* prevalence ratio, *CI* confidence interval.

[a]*p* value for interaction term between wave and characteristic according to a two-tailed chi-square type 3 analysis. Significant values indicate substantial changes in PRs across waves for a specific characteristic. Full list of *p* values in Supplementary Table 5.

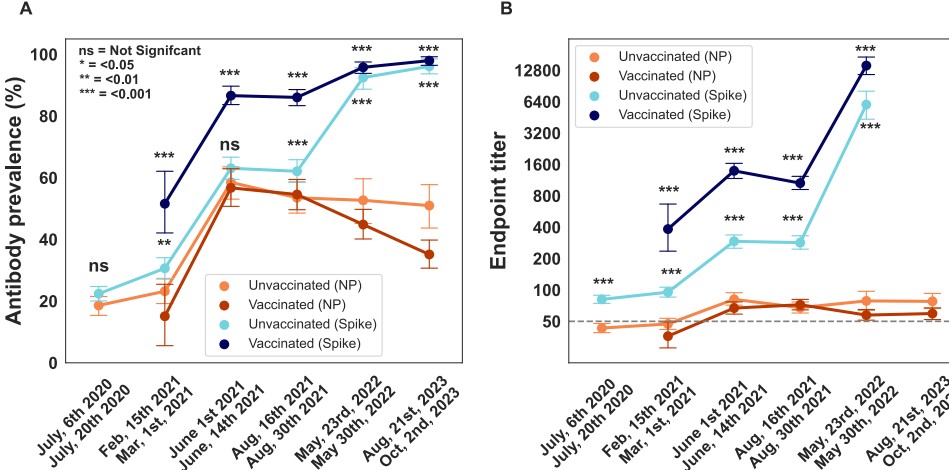

**Fig. 3 | SARS-CoV-2 nucleoprotein (NP) and spike (S) antibody prevalence and titers over six sampling periods (2-week duration) through the course of the pandemic.** NP and S antibody prevalence (**A**) and titers (**B**) in patients attending Mount Sinai Hospital in New York City are shown. Data stratified by vaccination status, assessed at the time of sample collection. Sampling time points: July 6–20th, 2020 (*n* = 661); February 15th to March 1st, 2021(*n* = 497); June 1st to June 14th, 2021(*n* = 610); August 16th to August 30th, 2021 (*n* = 758); May 23rd to May 30th, 2022 (*n* = 592); Aug 21st to Oct 2nd, 2023 (*n* = 595). Antibody prevalence (%) with 95% confidence intervals is shown in **A**. Geometric mean of endpoint titers plus 95% confidence intervals are shown in **B**. Samples were run in duplicate under our 2-step ELISA protocol. Limit of detection (LoD) is indicated by the horizontal dotted line. Statistically significant differences between NP and S values are indicated as follows: ns, not significant; * <0.05, ** <0.01, *** <0.001. All exact values shown within the figure are provided in Supplemental Table 14. A 2-tailed Chi-squared test for antibody prevalence was conducted between spike and NP positivity within the vaccinated and unvaccinated subgroup, and a 2-tailed T-test for the same targets (spike vs. NP within vaccinated and unvaccinated subgroups) was conducted for endpoint titer. All exact *p* values are shown in Supplemental Table 15.

## Methods

### Study participants and human samples

Sample collection and testing started on February 9th, 2020, and ended on July 18th, 2022. Follow-up testing started on August 21st 2023 and ended on October 2nd 2023. Collection was performed on a weekly basis (*n* = 608/week on average) and in a blinded manner.

Plasma was collected from ethylenediaminetetraacetic acid (EDTA)-treated blood specimens that remained from standard-of-care testing at the Mount Sinai Hospital (MSH) Blood Bank. Samples were sorted according to collection week, visit location, and practice, including OB/GYN, labor and deliveries, oncology, surgery, cardiology, emergency department, and other related hospital admissions. Up to

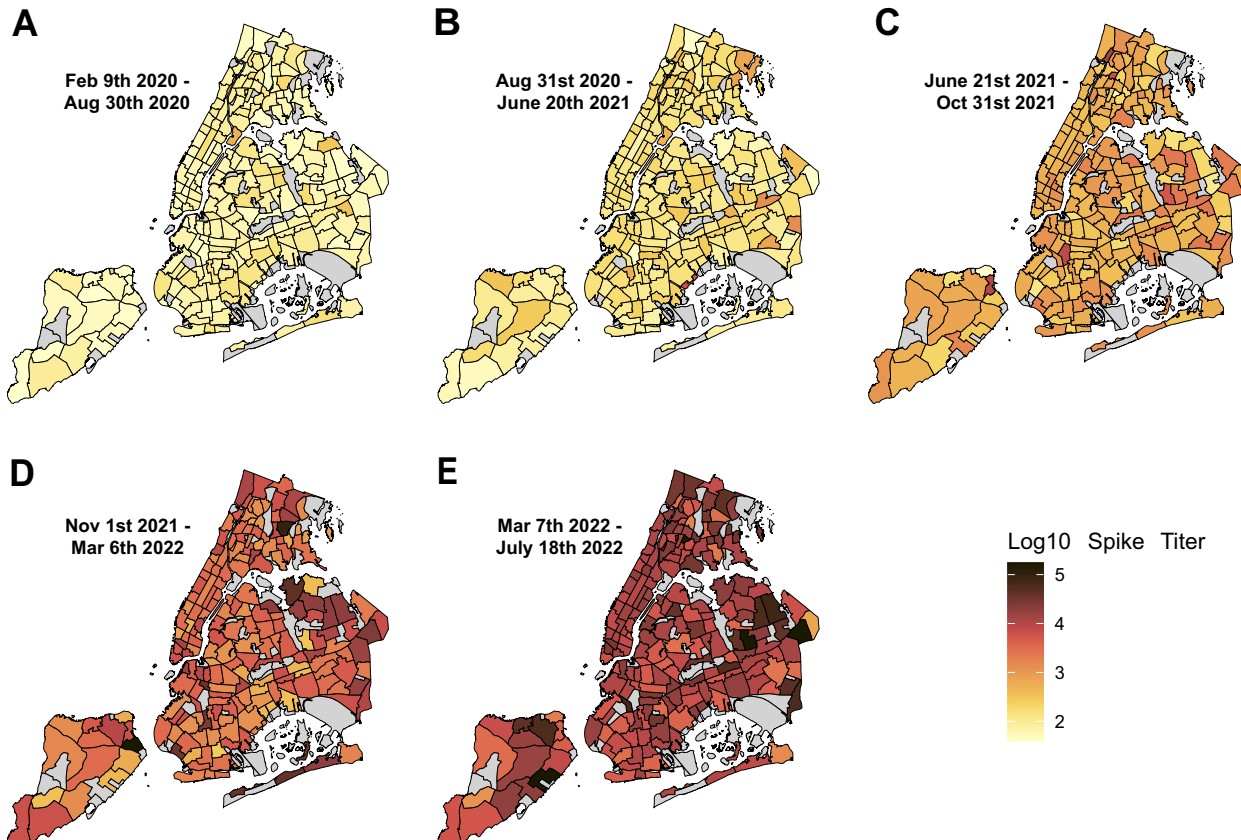

**Fig. 4 | Geographical distribution of SARS-CoV-2 spike antibody titers during five epidemiological waves of COVID-19 in residents of New York City (NYC).** SARS-CoV-2 spike IgG titer measured in patients from the five boroughs of NYC (Manhattan, Brooklyn, Queens, The Bronx, and Staten Island) attending to a Mount Sinai Hospital in NYC are shown (latitude and longitude: 40.7128° N, 74.0060° W). Five epidemiological waves corresponding to successive peaks of COVID-19 incidence are shown. **A** Wave 1 (February 9th to August 30th, 2020). **B** Wave 2 (August 31st, 2020, to June 20th, 2021). **C** Wave 3 (June 21st to October 31st, 2021). **D** Wave 4 (November 1st, 2021, to March 6th, 2022). **E** Wave 5 (March 7th to July 18th, 2022). Antibody titer is expressed as log10 geometric mean titer. Areas with l<10 specimens are shaded gray. Color gradient depicts range of antibody titers.

152 specimens at each location in a defined collection week were randomly selected. Our main analyses and figures include unweighted data, to reflect the serosurveillance system's aim of being able to provide immunological information in real-time. As a sensitivity analysis, we have also included a weighted analysis of our main seroprevalence results. Weights were based on Census 2022 age and gender estimates of the New York City population.

During sample aliquoting, specimens were captured into a coded database with randomly assigned identifiers and verified patient categories. Additional specimens obtained from the same patient within the same week were not considered for analysis. Samples collected 7 days apart or more from the first specimen collected were considered independent samples of the population. Typically, 67-314 specimens were selected weekly from the 'routine care' group (OB/GYN, labor and deliveries, oncology, surgery, and cardiology) and 67 to 314 samples from the 'urgent care' group consisting of specimens from patient visits to the emergency department and urgent care. Some individuals had a PCR test to diagnose SARS-CoV-2 infection. The analyses of samples collected during the week ending on February 9th 2020 to July 5th 2020 were reported previously[8] and detailed description of serological methods can be found there. Briefly, serological testing was based on an initial screening step against the recombinant receptor binding domain (RBD) of SARS-CoV-2 followed by titration of positive samples for S[28,29]. RBD and S used here were derived from the ancestral strain of SARS-CoV-2. We incorporate all samples with additional

demographics data in the current analysis in order to describe seroprevalence and antibody titers since the beginning of the pandemic. We stratify our analyses by 'urgent care' vs 'routine care' groups to capture the fact that access to non-urgent medical care was limited at the beginning of the pandemic[30].

Specimen metadata included information on the individuals' age, sex, race/ethnicity, insurance status, and whether they had received a COVID-19 vaccination prior to specimen collection. Due to the low number of children in our study, we limited our main analysis to adults and divided them into 18–44 years, 45–64 years, and 65+ years age groups. Information on race was based on self-reported categories (e.g., Asian, Pacific Islander, Black, White, Other, and unknown). Information on ethnicity (e.g., Hispanic or non-Hispanic) was not consistently available and, thus, not included in our analysis. Sex was based on self-reporting. Medicaid and Medicare were collapsed into a public insurance category. All data used in this study was anonymized and recoded through the use of an honest broker following local regulations.

**Epidemiological waves**
We categorized samples by time into 5 epidemiological waves, that correspond to successive peaks in COVID-19 incidence in NYC: Wave 1 (February 9th to August 30th, 2020; ancestral SARS-CoV-2 and D614G mutant circulating), Wave 2 (August 31st, 2020, to June 20th, 2021; Iota (B.1.529) and Alpha (B.1.1.7) circulating), Wave 3 (June 21st to October 31st, 2021; Delta (B.1.617.2) circulating), Wave 4 (November 1st, 2021, to

March 6th, 2022; Delta (B.1.617.2) and Omicron BA.1 (B.1.1.529.1) circulating), and Wave 5 (March 7th to July 18th, 2022; Omicron BA.2 (B.1.1.529.2) and Omicron BA.5 (B.1.1.529.5) circulating). Epidemiological waves and circulating variants are depicted in Fig. 1A, B, respectively. The follow-up period (August 21st 2023 to October 2nd 2023), was not associated to a specific wave timeframe, and was included to assess the maintenance of seroprevalence and durability of spike binding antibody titers.

### Geographical maps
The NTA map created by the Department of City Planning and sourced from NYC OpenData was used as the base geometry for our data analyses. NTAs are neighborhoods that consist of multiple Census tracts, have on average a population of 42,000, and represent the smallest geographical unit that we were able to obtain under our protocol. For each NTA, the average of the common logarithm of spike titer values was calculated and plotted as a shaded map using the ggplot2 and sf packages in R. For plots showing progression over time, slices were taken from each data range and plotted as separate facets.

### Statistical analysis
We describe the distribution of results graphically and through proportions, stratified by 'urgent care' vs 'routine care' groups, with any denominator being the total number of individuals with tested sera. Visualizations of antibody prevalence and titers were generated using Matplotlib (3.7.2), Seaborn (0.11.2), Pandas (1.5.3), Numpy (1.26.0), and Scipy (1.11.3) packages within python 3.11.5. Geographical charts, and live data explorer were generated using R version 4.3.2, with the packages sf (1.0.14), DT (0.31), zoo (1.8.12), Hmisc (5.1.1), shiny (1.8.0), scico (1.5.0), ggpubr (0.6.0), bsplus (0.1.4), forcats (1.0.0), ggplot2 (3.4.4), ggbreak (0.1.2), patchwork (1.1.3), lubridate (1.9.3), hrbrthemes (0.8.0), data.table (1.14.10), shinyWidgets (0.8.0), RColorBrewer (1.1.3), shinydashboard (0.7.2), and shinycssloaders (1.0.0).

Based on a priori considerations, we include age, sex, race/ethnicity, and insurance status in our multivariable models: age representing differential humoral response to infection and increased risk of severe disease[31,32] and therefore a marker of potentially enhanced social distancing behaviors[33], sex-linked to health seeking behaviors[34] as well as gynecological appointments, and race and insurance status measures of poverty, health-seeking behaviors, and exposure to SARS-CoV-2 through a variety of mechanisms[35].

Differences in seropositivity by demographic group over time were assessed through multivariable, Poisson regression models that included robust variance estimates[36]. These models output prevalence ratios (PR) and 95% confidence intervals (CI), which were interpreted relative to a reference group. These models estimated the direct effect of age, sex, race/ethnicity, and insurance status on seropositivity. These models were initially stratified by epidemiological wave. To estimate changes in seropositivity by demographic group across waves, we also specified a larger model, where an interaction term was placed between wave and every demographic variable. The $p$ value from this interaction term indicates if there has been a significant shift in differences across wave, for a given demographic variable.

We conduct several sensitivity analyses. In a separate analysis, we also include vaccination, but limit the timing to data collected after January 11, 2021, when the vaccine started becoming available to certain age groups in the general population[37]. In a third set of models, we exclude individuals 65+ based on their ability to access Medicare. The regression models were implanted in SAS version 9.4 (SAS Institute, Cary, NC, USA). We used an alpha level of 0.05 to assess significance. All analyses were either stratified by care setting (routine vs urgent care) or were limited to those in the routine care group.

### Ethical approval
The study protocol HS 20-00308 was reviewed by the Mount Sinai Health System Institutional Review Board, Icahn School of Medicine at Mount Sinai, and it was exempt from human research as defined by regulations of the Department of Health and Human Services (45 CFR 46. 104). The need for written informed consent for patient samples was waived.

### Reporting summary
Further information on research design is available in the Nature Portfolio Reporting Summary linked to this article.

## Data availability
All data are available under ImmPort accession # SDY2491 at https://www.immport.org/home.

## Code availability
Code is available at https://doi.org/10.6084/m9.figshare.25962448.

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

## Acknowledgements

This effort was supported by the Serological Sciences Network (SeroNet) in part with Federal funds from the National Cancer Institute, National Institutes of Health, under Contract No. 75N91019D00024, Task Order No. 75N91021F00001 (to F.K., V.S.). The content of this publication does not necessarily reflect the views or policies of the Department of Health and Human Services, nor does mention of trade names, commercial products, or organizations imply endorsement by the U.S. Government. This work was also partially funded by the Centers of Excellence for Influenza Research and Surveillance (CEIRS, contract # HHSN272201400008C to F.K.), the Centers of Excellence for Influenza Research and Response (CEIRR, contract # 75N93021C00014 to F.K., V.S.), by the Collaborative Influenza Vaccine Innovation Centers (CIVICs contract # 75N93019C00051 to F.K., V.S., A.G.) and by institutional funds.

## Author contributions

F.K., V.S., H.v.B., A.G., E.M.S., and J.M.C. conceived and designed the study. H.v.B. wrote and maintained the Institutional Review Board protocol. J.T., A.R., D.B., G.S., S.M., M.F., T.Y., and L.S. performed the serological assays. J.M.C. and F.K. supervised serological assays. D.B., S.M., and B.M. collected, organized, and aliquoted plasma samples. A.S.G.R. provided data on SARS-CoV-2 circulating lineages detected at Mount Sinai. G.S. produced and purified recombinant proteins. J.M.C., J.T., B.M., and A.L.W. collected and analyzed the data. J.M.C., B.M., and D.F. generated figures. J.M.C. and A.L.W. wrote the original manuscript drafts. All authors edited and approved the manuscript.

## Competing interests

The Icahn School of Medicine at Mount Sinai has filed patent applications relating to SARS-CoV-2 serological assays and NDV-based SARS-CoV-2 vaccines which list Florian Krammer as co-inventor. Dr. Simon is listed on the SARS-CoV-2 serological assays patent. Mount Sinai has spun out a company, Kantaro, to market serological tests for SARS-CoV-2. Dr. Krammer has consulted for Merck, Seqirus, CureVac, GSK and Pfizer in the past and is currently consulting for Gritstone Bio, 3rd Rock Ventures, and Avimex, and he is a co-founder and scientific advisory board member of CastleVax. The Krammer laboratory has been collaborating with Pfizer on animal models for SARS-CoV-2. The remaining authors declare no competing interests.

## Additional information

## PARIS study group

Fatima Amanat[1], Guha Asthagiri Arunkumar[1], Christina Capuano[1], Jordan Ehrenhaus[1], Shelcie Fabre[1],
Matthew M. Hernandez[1], Kaijun Jiang[1], Brian Lerman[1], Meagan McMahon[1], Daniel Stadlbauer[1], Jessica Tan[1], Catherine Teo[1]
& Kathryn Twyman[11]

[11]The Mount Sinai Data Office, Mount Sinai Health System, New York, NY, USA.

