## [Peer Review File · Nature Communications]

SARS-CoV-2 serosurvey across multiple waves of the COVID-19 pandemic in New York City between 2020-2023REVIEWER COMMENTS

Reviewer #1 (Remarks to the Author):

The manuscript details a cross-sectional study of seroprevalence in New York City throughout the pandemic using hospital specimens.

The manuscript is well written, but I have questions about the study design and analysis, which relate to the overall relevance of the paper.

Who is the population of interest for the study? On line 303 in the discussion when comparing these study results to that from NYC DOHMH the text reads, "This could reflect differences in the assays used or differences in the study populations." Table 1 shows that the demographics of the urgent care population are distinctly different than the routine visit population. If Table 1 were further decomposed by wave, I believe we would see additional heterogeneity over time for both populations. For this reason, it feels inappropriate to simply aggregate all samples and perform any analysis (e.g. Table 2). I find it extremely difficult to think about what population we are learning about from any combined analysis. In addition, there is a substantial risk of making misleading conclusions due to Simpson's paradox.

I find the results about vaccination rates on lines 197-199 to be especially suspect. The text discusses differences in vaccination by racial subgroups and points the reader to Supplemental Table 5. This is a table summarizing vaccination rates for all patients (both routine visit and urgent care visit) in each wave. It seems very plausible that in the waves of the pandemic after vaccines are available unvaccinated individuals are more represented in the urgent care visit group (since it is more likely they contracted and developed COVID-19). It is very difficult to interpret Supplemental Table 5 for this combined group of patients.

It would seem the routine visit samples alone could be reweighted to make seroprevalence inferences about the population of New York City. This latter population is arguably a population of greater interest and relevance for public health. The data from the children < 18 could also be incorporated or the target population could remain adults. Nevertheless, this would demonstrate how hospital-based serosurveys could be used to gain more general insights about a community.

This paper is not the first to do a cross-section serosurvey of a population. Thus, I am left wondering what the unique contributions are of this work. A greater literature review of other existing studies and the specific relevance of this work needs to be better articulated in the introduction and conclusion.

Minor Comments and Suggestions:

- Figure 1: It would be helpful if annotations were added above the shaded regions stating "Wave 1", "Wave 2", ..., "Follow up". This could be added to just panel A or all panels.
- Figure 1 E-F are missing y-axis labels.
- Line 145: The average of the common logarithm of spike titer values was calculated. Is the distribution of the logarithm of spike titer values roughly symmetric? If not, the median would be a better summary measure.
- In numerous places, the text uses "Black" to identify a group of individuals but "Black" is not a demographic category listed in Table 1 and is not the same as African American. See lines 226, 235, and others.
- Some mentions of Tables and Figures are bolded and others are not.
- Supplemental Tables are referenced out of order in the text.

Reviewer #2 (Remarks to the Author):

This excellent study reports the results of a repeat cross sectional study whose purpose was to monitor population immunity over time, through detection of anti-Spike and anti-Nucleocapsid antibodies from testing of leftover serum from routine and urgent care at a single large hospital in New York City, Mt Sinai Hospital. The research is original, sound, and adds to our knowledge. This study is a follow up to an earlier study which reported seroprevalence during Feb 2020 to July 2022; those results are included here in addition to the follow up data from Aug 2023 to Oct 2023. Metadata including vaccination status were available, which adds to the strength of this study, as it allows tracking of patterns among vaccinated and unvaccinated. Other metadata include age, race, and sex, which allow for important description of population differences in seroprevalence over time, particularly among diverse racial\ethnic groups.

The most important finding from this study is that current population immunity is greater than 90% in this population among both vaccinated and unvaccinated persons. The higher seroprevalence in the vaccinated group prior to the third wave is also noteworthy. By the third wave, most were immune. Although the results are from a single hospital, two factors support the representativeness of this sample- first, the large sample size of more than 55,000 persons, and second, the finding that results are similar despite the type of care setting, urgent vs. routine care. Indeed, the fact that 51% did not have private insurance suggests that this sample is representative of vulnerable, low income populations in NYC

Minor comments

- 1) Could the authors please explain their use or rationale for Neighborhood Tabulation areas?
- 2) Clarification- was the antibody prevalence calculated using as a denominator all people seen at the hospital who resided in a given geographic area, or was the denominator the persons seen at the hospital and whose serum was tested who lived in that geographic area?
- 3) The manuscript has many figures. I would suggest the authors identify the most important, and put several of the less critical figures in an online supplement.

We thank the reviewers for their thoughtful comments and have responded to them below point by point. Line numbers refer to the redlined pdf version of the manuscript.

Reviewer #1 (Remarks to the Author):

The manuscript details a cross-sectional study of seroprevalence in New York City throughout the pandemic using hospital specimens.

The manuscript is well written, but I have questions about the study design and analysis, which relate to the overall relevance of the paper.

Who is the population of interest for the study? On line 303 in the discussion when comparing these study results to that from NYC DOHMH the text reads, ???This could reflect differences in the assays used or differences in the study populations.??? Table 1 shows that the demographics of the urgent care population are distinctly different than the routine visit population. If Table 1 were further decomposed by wave, I believe we would see additional heterogeneity over time for both populations. For this reason, it feels inappropriate to simply aggregate all samples and perform any analysis (e.g. Table 2). I find it extremely difficult to think about what population we are learning about from any combined analysis. In addition, there is a substantial risk of making misleading conclusions due to Simpson???'s paradox.

Authors' response: We agree with the reviewer that care should be taken to interpret results from the urgent care group. We apologize for any confusion on this matter, but we affirm that all results are already stratified by care setting. We have made this clear in the text. When we only present one set of results, it is because we are not showing data from the urgent care group. We now state this in the methods (line 357-358):

“All analyses were either stratified by care setting (routine vs urgent care), or were limited to those in the routine care group.”

We have relabeled Table 2 in the text to make this clear.

“Table 2. Multivariable Poisson regression of spike protein seropositivity among those receiving routine care in a SARS-CoV-2 serosurveillance system, New York City.”

I find the results about vaccination rates on lines 197-199 to be especially suspect. The text discusses differences in vaccination by racial subgroups and points the reader to Supplemental Table 5. This is a table summarizing vaccination rates for all patients (both routine visit and urgent care visit) in each wave. It seems very plausible that in the waves of the pandemic after vaccines are available unvaccinated individuals are more represented in the urgent care visit group (since it is more likely they contracted and developed COVID-19). It is very difficult to interpret Supplemental Table 5 for this combined group of patients.

Authors’ response: From re-ordering of the tables this is now Supplementary Table 2. We agree that the unvaccinated could theoretically be more represented in the urgent care group. Our analysis never included them but we were not clear in our Table titles. Supplementary Table 2 is renamed the following (and other tables similarly renamed):

Supplementary Table 2. COVID-19 vaccination coverage over time among those receiving routine care in a SARS-CoV-2 serosurveillance system, New York City.

It would seem the routine visit samples alone could be reweighted to make seroprevalence inferences about the population of New York City. This latter population is arguably a population of greater interest and relevance for public health. The data from the children < 18 could also be incorporated or the target population could remain adults. Nevertheless, this would demonstrate how hospital-based serosurveys could be used to gain more general insights about a community.

Authors’ response: We appreciate your insights. As previously mentioned, our sample in most analyses is limited to the routine care patients.

We note that some basic data from children is shown in Supplementary Table 1. Unfortunately, we did not have an adequate sample size to conduct additional multivariable analyses.

We had also discussed weighting of the sample. Ultimately, we decided not to, as we would like to position this paper as an example of how surveillance systems in the future could prospectively assess the impact of a pandemic using serology. Statistical weighting requires a retrospective analysis, as a researcher would need to decide what dates of surveillance data collection would be included in the weighting. Moreover, a future pandemic could be associated with substantial, and demographically differential migration patterns, which could limit the utility of using publicly available population numbers as a standard population size.

This paper is not the first to do a cross-section serosurvey of a population. Thus, I am left wondering what the unique contributions are of this work. A greater literature review of other existing studies and the specific relevance of this work needs to be better articulated in the introduction and conclusion.

Authors' response: We appreciate the reviewer's comments and believe the uniqueness of our study lies in the use of a hospital-based sero-monitoring system, the length of the study (2020-2023), and its application to examining demographic differences in seropositivity over time and in conjunction with the vaccination roll-out.

We have added more context to this study in the introduction (line 87-92):

“Hospital-based sero-monitoring for SARS-CoV-2 provides an estimate of the level of immunity in the population and could provide a blue-print for how to conduct such surveillance for a future pandemic. This sero-monitoring could provide important information about the course of the pandemic, who are high risk groups, and the persistence of disparities even after the introduction of population-level interventions. Existing case reporting systems for COVID-19 could be limited by changing testing availability and behaviors over time.”

We have added this to the discussion (line 242-248):

“Several sero-monitoring programs have been implemented worldwide in response to the SARS-CoV-2 pandemic. These systems have been designed for various purposes, including to estimate antibody seropositivity ³², to monitor development of community immunity ³³, and to examine occupational risks ³⁴. Community-based seroprevalence

studies (e.g. Kshatri et al. 35) have several limitations, including costs of conducting field studies and non-response, which could be differential based on important demographic groups. Hospital-based sero-monitoring systems, as we describe, can leverage existing laboratory infrastructure in a low-cost manner.”

Minor Comments and Suggestions:

- Figure 1: It would be helpful if annotations were added above the shaded regions stating ???Wave 1???, ???Wave 2???,???,???Follow up???. This could be added to just panel A or all panels.

Authors’ response: We appreciate the helpful comment and have added annotations to panel A.

- Figure 1 E-F are missing y-axis labels.

Authors’ response: Thank you. We have added the missing labels.

- Line 145: The average of the common logarithm of spike titer values was calculated. Is the distribution of the logarithm of spike titer values roughly symmetric? If not, the median would be a better summary measure.

Authors’ response: Yes, we believe that the logarithm of spike titers is roughly symmetrical. See histogram below:

-In numerous places, the text uses ???Black??? to identify a group of individuals but ???Black??? is not a demographic category listed in Table 1 and is not the same as African American. See lines 226, 235, and others.

Authors' response: For consistency, we have changed these all to "Black."

- Some mentions of Tables and Figures are bolded and others are not.

Authors' response: We have bolded these now and defer to the journal copy editor as to the required format.

- Supplemental Tables are referenced out of order in the text.

Authors' response: We have re-ordered tables and figures.

Reviewer #2 (Remarks to the Author):

This excellent study reports the results of a repeat cross sectional study whose purpose was to monitor population immunity over time, through detection of anti-Spike and anti-

Nucleocapsid antibodies from testing of leftover serum from routine and urgent care at a single large hospital in New York City, Mt Sinai Hospital. The research is original, sound, and adds to our knowledge. This study is a follow up to an earlier study which reported seroprevalence during Feb 2020 to July 2022; those results are included here in addition to the follow up data from Aug 2023 to Oct 2023. Metadata including vaccination status were available, which adds to the strength of this study, as it allows tracking of patterns among vaccinated and unvaccinated. Other metadata include age, race, and sex, which allow for important description of population differences in seroprevalence over time, particularly among diverse racial\ethnic groups. The most important finding from this study is that current population immunity is greater than 90% in this population among both vaccinated and unvaccinated persons. The higher seroprevalence in the vaccinated group prior to the third wave is also noteworthy. By the third wave, most were immune. Although the results are from a single hospital, two factors support the representativeness of this sample- first, the large sample size of more than 55,000 persons, and second, the finding that results are similar despite the type of care setting, urgent vs. routine care. Indeed, the fact that 51% did not have private insurance suggests that this sample is representative of vulnerable, low income populations in NYC

Authors' response: We appreciate the reviewer's comments and their suggestions for revising the manuscript.

Minor comments

1) Could the authors please explain their use or rationale for Neighborhood Tabulation areas?

Authors' response: We have added more detail in the methods (line 317-323):

“Geographical maps. The neighborhood tabulation area (NTA) map created by the Department of City Planning and sourced from NYC OpenData was used as the base geometry for our data analyses. NTAs are neighborhoods that consist of multiple Census tracts, have on average a population of 42,000, and represent the smallest geographical unit that we were able to obtain under our protocol. For each NTA, the average of the common logarithm of spike titer values was calculated and plotted as a shaded map using

the ggplot2 and sf packages in R. For plots showing progression over time, slices were taken from each data range and plotted as separate facets. “

2) Clarification- was the antibody prevalence calculated using as a denominator all people seen at the hospital who resided in a given geographic area, or was the denominator the persons seen at the hospital and whose serum was tested who lived in that geographic area?

Authors' response: The latter - denominator was only based off of tested samples. We clarify this in the methods (line 325-327):

“We describe the distribution of results graphically and through proportions, stratified by ‘urgent care’ vs ‘routine care’ groups, with any denominator being the total number of individuals with tested sera.”

3) The manuscript has many figures. I would suggest the authors identify the most important, and put several of the less critical figures in an online supplement.

Authors' response: We have removed two figures to the supplementary appendix. We appreciate the suggestion.

REVIEWER COMMENTS

Reviewer #1 (Remarks to the Author):

I appreciate the authors' efforts in preparing a revision, however, I still have a couple of serious concerns with the paper in its current condition.

It is helpful to know that the routine visit population was the basis for most of the analyses. That said, a population of interest should be clearly defined for this study. It seems conceivable that either the population of interest could be defined as (1) the population of NYC, or (2) the population of `routine care` patients pre-pandemic. Lines 96-97 indicate the `routine care` group resembles the general population and cites a 2021 paper, however the study at hand contains data through October 2023. It does not appear a detailed investigation was performed to assess how the `routine care` group changed over time. At a minimum (and as suggested in my first review), Table 1 should be presented broken down by wave so that the reader can better understand how the demographics of the `routine care` (and in-patient) changed over time. Assuming the characteristics of the `routine care` group of patients did change over time, as stated in my first review, the samples within each wave should be reweighted back to the population of interest. For Figures 1 & 2, this weighting could be performed using data from a single week or a large window could be used. If the demographics changed little over time, this reweighting should affect results minimally. However, this ensures meaningful comparisons can be made over time in these figures.

Regarding the statistical analysis, the text (pg 13, lines 349-350) and footnote on Table 2 note that to determine whether patterns of seropositivity change across waves by demographic group a larger model was fit with interaction terms between wave and every demographic variable. A single p-value is given for each categorical demographic variable in the last column of Table 2 representing the results. I have a couple of questions about this:

1. Was wave included as a main effect in this interaction model? It is standard to include a main effect any time an interaction term is included.

2. How was a single p-value obtained for each interaction between wave and demographic variable? This is confusing as an interaction term between wave and age, for example, would result in 8 interaction parameters, each of which would have its own p-value in the Poisson regression. (Note: Eight parameters arise assuming 2 age levels x 4 waves, besides the baseline levels, where main effects are included for wave and age.) A likelihood ratio test could be used to compare nested models, but these methods are not mentioned in the methods section leaving me curious as to what the authors did.

Reviewer #1 (Remarks to the Author):

I appreciate the authors' efforts in preparing a revision, however, I still have a couple of serious concerns with the paper in its current condition.

We thank the reviewer for the critical comments that have helped us to improve the content of our manuscript. We have worked to adequately address all the comments.

It is helpful to know that the routine visit population was the basis for most of the analyses. That said, a population of interest should be clearly defined for this study. It seems conceivable that either the population of interest could be defined as (1) the population of NYC, or (2) the population of `routine care` patients pre-pandemic. Lines 96-97 indicate the `routine care` group resembles the general population and cites a 2021 paper, however the study at hand contains data through October 2023. It does not appear a detailed investigation was performed to assess how the `routine care` group changed over time. At a minimum (and as suggested in my first review), Table 1 should be presented broken down by wave so that the reader can better understand how the demographics of the `routine care` (and in-patient) changed over time.

Authors' response: We now include a new table where we break down demographic characteristics of those in the routine care program by epidemiological wave: 'Supplementary Table 3. Demographic characteristics by epidemiological wave among those receiving routine care in a SARS-CoV-2 serosurveillance system, New York City'. We have also indicated in the results section (page 5 line 115): "Demographic characteristics did not substantially vary by wave (Supplementary Table 3)".

Assuming the characteristics of the `routine care` group of patients did change over time, as stated in my first review, the samples within each wave should be reweighted back to the population of interest. For Figures 1 & 2, this weighting could be performed using data from a single week or a large window could be used. If the demographics changed little over time, this reweighting should affect results minimally. However, this ensures meaningful comparisons can be made over time in these figures.

Authors' response: We now include supplementary figures 1-2, which include data that have been weighted to reflect Census 2022 estimates for New York City. We have also added to the methods (page 10 line 277): "Our main analyses and figures include unweighted data, to reflect the serosurveillance system's aim of being able to provide immunological information in real-time. As a sensitivity analysis, we have also included

a weighted analysis of our main seroprevalence results. Weights were based on Census 2022 age and gender estimates of the New York City population.”

In the methods we write (page 5 line 137): “A sensitivity analysis of trends in seropositivity over time, using data weighted to Census 2022 age and gender estimates, did not reveal any substantively different results (**Supplementary Figures 1-2**).”

Regarding the statistical analysis, the text (pg 13, lines 349-350) and footnote on Table 2 note that to determine whether patterns of seropositivity change across waves by demographic group a larger model was fit with interaction terms between wave and every demographic variable. A single p-value is given for each categorical demographic variable in the last column of Table 2 representing the results. I have a couple of questions about this:

1. Was wave included as a main effect in this interaction model? It is standard to include a main effect any time an interaction term is included.

2. How was a single p-value obtained for each interaction between wave and demographic variable? This is confusing as an interaction term between wave and age, for example, would result in 8 interaction parameters, each of which would have its own p-value in the Poisson regression. (Note: Eight parameters arise assuming 2 age levels x 4 waves, besides the baseline levels, where main effects are included for wave and age.) A likelihood ratio test could be used to compare nested models, but these methods are not mentioned in the methods section leaving me curious as to what the authors did.

Authors’ response: We now include Supplementary Table 5, which displays the results of the full model. The p-value in Table 2 is for a “Type 3” analysis, which is a type of likelihood ratio test. In our Supplementary Table 5, we also include category-specific P-values.

In the footnote for table 2 we now write “according to a type 3 analysis” as a further clarification.

Supplementary Table 5. Full parameter estimates from a multivariable Poisson regression of spike protein seropositivity among those receiving routine care in a SARS-CoV-2 serosurveillance system, New York City.

We reference this in the results (page 5 line 138): “The multivariable Poisson regression model of spike protein seropositivity reveals several trends within demographic group and across time (**Table 2, Supplementary Table 5**).”